# Need for Uniform Definitions in Newborn Screening for SCID: The Next Challenge for Screeners and Immunologists

**DOI:** 10.3390/ijns7030052

**Published:** 2021-08-06

**Authors:** Mirjam van der Burg

**Affiliations:** Department of Pediatrics, Laboratory of Pediatric Immunology, Willem-Alexander Children’s Hospital, Leiden University Medical Center, Albinusdreef 2, 2333 ZA Leiden, The Netherlands; m.van_der_burg@lumc.nl; Tel.:+31-71-526-4976

**Keywords:** severe combined immunodeficiency, TREC, classification, case definition

## Abstract

During the ISNS meeting “Newborn Screening for SCID ‘State of the Art’” on 26 and 27 January 2021, the topic of case definitions and related issues were discussed. There is currently a lack of uniform definitions and therefore a lack of uniform registration of screen-positive cases. This severely hampers the comparison of outcomes of different screening programs and the exchange of experiences gained by the different countries performing SCID screening, which is essential to improve screening programs. In this letter, I outline the current situation and indicate the need for uniform definitions and classification, which in my view needs to be a joined effort of screeners and immunologists.

## 1. Introduction

Coming from the field of diagnostics and research on severe combined immunodeficiency, I entered the world of newborn screening several years ago. The moment I actually realized that the field of SCID diagnostics and research is a different world than the field of newborn screening was during a visit to the laboratory of Anne Marie Comeau in Massachusetts. She was running the SCID screening laboratory and showed me her in-house TREC assay procedure and we discussed her wish to set up KREC screening. With great enthusiasm, she explained the logistics of the screening process, and at a certain moment she said: “I am a newborn screener”. Then it became clear to me: this is a different profession.

I think that this difference explains the tension one can feel between population-based screening on one hand and diagnostics of a child with a clinical suspicion of SCID on the other hand. It is this tension that makes it a challenge for both screeners and immunologists (clinical and laboratory) to ensure the most optimal screening program with the most optimal follow-up diagnostics in order to reach the most optimal clinical care for SCID. This challenge is identical for all screening programs, but I think SCID screening has its own unique challenges.

## 2. Balance between Identification of SCID Patients and Incidental Findings

Results from pilot studies and implemented screening programs show that SCID as a target disease can be identified very efficiently with TREC screening [1]. Thus far, there are no reports of missed SCID cases, illustrating the power of this approach [2]. However, TREC screening is accompanied by many incidental findings, i.e., patients without SCID but with low numbers of T-cells for other reasons [3]. The number of these non-SCID cases is much higher than the number of SCID cases. For some of these patients, there is a clear clinical benefit because they receive prophylaxis, adaptation of the vaccination scheme or even a hematopoietic stem cell transplantation [2]. From a screening perspective, one could argue that this is not the reason for screening and incidental findings should be avoided as much as possible. Opinions on this issue vary substantially, especially between different countries. Another point that should be taken into consideration is the emotional impact of an abnormal screening result on parents, irrespective of the final outcome of the diagnostic process [4]. Altogether, there is a need to find the optimal balance between the identification of SCID patients and the pickup rate of incidental findings.

## 3. Need for Uniform Definitions and Classification

Parameters that directly influence the number and type of referrals are [1] the used TREC assay in combination with the cut-off value, [2] the screening algorithm, [3] the referral policies including the policy for pre-terms, and [4] the potential use of second tier tests and genetics. These parameters vary per screening program. At this moment, it is difficult to compare the outcome of the different screening programs because there is no uniform registration of referrals (i.e., the screen-positive cases). There are several diagnostic guidelines for SCID, which all slightly differ in the exact definition and may or may not be uniform in the definition of, e.g., leaky SCID and atypical SCID with or without a genetic diagnosis. The guidelines from the Primary Immune Deficiency Treatment Consortium (PIDTC) are most widely used and may be regarded as the standard [5].

However, there is no international consensus about which definitions are used for reporting screening outcomes, especially for the non-SCID cases, incidental findings and how exactly false-positives are defined. This is illustrated in Table 1 in which the disease categories of six publications are summarized. In this table I categorized them into SCID, non-SCID, preterm and false positive, which are broad categories. In the six publications, several terms are used such as T-cell impairment syndrome, idiopathic lymphocytopenia, and syndromes. In addition, some studies report on pre-terms and false-positives, while other studies do not. This lack of uniform registration severely hampers the comparison of different screening programs, which is essential to improve SCID screening. I would suggest that the disease categories need to be clearly and precisely defined, because “SCID” and “non-SCID” lack too much clinically relevant detail. I am convinced that at this moment, we therefore do not benefit enough form the experience gained by the different countries performing SCID screening.

## 4. Joined Forces of Screeners and Immunologists

Fifteen years after TREC-based screening has become available, it has been convincingly shown that the early identification of SCID babies allows prompt corrective treatment with improved overall survival. However, a screening program is broader than only the diagnosis of these SCID patients because it concerns a public health instrument in which parents of children without the “target disease” are also confronted with an abnormal screening result of their child in the first week of life. I think that screeners and immunologists need to join forces to define uniform definitions and a classification system to register all screen-positive children that allow the comparison of the different screening programs, and together make a policy for an optimal screening program. This will be a continuous process because there will be new developments in analytical possibilities that need to be carefully considered together with ethical aspects of screening. The recommendations for the uniform registry of case definitions need to be shared on a broad platform, including scientific conferences, newsletters and websites. In addition, they should be distributed via coordinating umbrella organizations in order to reach a broad public. To this end, I think organizations for immunologists or clinicians (ESID, CIS) as well as for NBS programs (ISNS, APHL, CLSI) should contribute to achieve the best possible program for SCID screening.

## Figures and Tables

**Table 1 IJNS-07-00052-t001:** Disorders considered screen positive cases in six newborn screening publications.

Disease Categories	Amatuni et al. [2]	Blom et al. [4]	Thomas et al. [6]	Knight et al. [7]	Argudo-Ramirez et al. [8]	Gans et al. [9]
SCID	SCID (typical, Omenn, Leaky)	SCID	SCID	Typical SCID	SCID	SCID
			leaky SCID and Omenn Syndrome	Leaky SCID or Omenn Syndrome		
			Variant SCID (T-cell lymphopenia)	Variant SCID/Idiopathic T-cell lymphopenia		
Non-SCID	Syndromes	Syndromes with T-cell impairment	T-cell impairment syndrome	Syndromes with T-cell lymphopenia	non-SCID lymphopenia	DiGeorge
						Other syndrome associated with lymphopenia
	Secondary	Secondary T-cell impairment	Secondary T-cell impairment	Secondary T-cell lymphopenia		
	Idopathic (until a diagnosis is made)	Idiopathic T-cell lymphocytopenia				Idiopathic lymphopenia
					transient Lymphopenia	
Preterm	Preterm		Preterm alone	Preterm Infants		
False Positive		False Positive			False positive	Immunocompetence

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
