# Peer review of "Need for Uniform Definitions in Newborn Screening for SCID: The Next Challenge for Screeners and Immunologists"

_2409-515X, 2021, doi:10.3390/ijns7030052_

Round 1

Reviewer 1 Report

Very nice letter with a high significance of content.

Uniform definitions and a classification system to register all screen-positive children could result in alterations of target disease definitions in screening programs. These alterations could be beneficial for patients with other T-cell related immunodeficiencies.  

Author Response

Thank you for the positive comment on my manuscript. I checked the manuscript for spelling.

Reviewer 2 Report

The evaluated manuscript represents a very important commentary on the field of SCID screening. The manuscript is presented by the author with long experience in the field of primary immunodeficiencies and, at the same time, with significant knowledge and experience in the field of SCID screening. In this reflection, the author points to the border between those two worlds, the field of clinical care for patients with SCID and the field of screening for those disorders. SCID screening has now been shown to be of great benefit to patients with SCID, but despite significant experience, there is still fragmentation between countries in the overall concept of screening, especially in the management of borderline or incidental findings. The manuscript focuses on this area and emphasizes the importance of cooperation between clinicians and screening specialists at the national level, but also at the international level, and calls for the unification of strategies and information sharing. In this respect, this commentary is very valuable and the manuscript is very high quality.

The only commentary on the text is the fact that the manuscript did not take the opportunity to suggest paths and strategies for the development of suggested international cooperation.

Minor comments -  Table 1 needs a graphic design to make the the message clear.

Author Response

I would like to thank the reviewer for the positive feedback.

He/she raised to important point that I addressed in the revised version:

  1. The only commentary on the text is the fact that the manuscript did not take the opportunity to suggest paths and strategies for the development of suggested international cooperation.

REPLY

I added the suggestion at the end of the last section. I recommended that both clinical and screening organisation should play an important role in this. I phrased it as follows:

The recommendations for uniform registry of case definitions need to be shared on a broad platform including scientific conferences, newsletters and websites. In addition they should be distributed via coordinating umbrella organizations in order to reach a broad public. To this end I think both organizations for immunologists or clinicians (ESID, CIS) as well as for NBS programs (ISNS, APHL, CLSI) should contribute to achieve the best possible program for SCID screening.

Minor comments -  Table 1 needs a graphic design to make the the message clear.

REPLY

I used the indicated style of the journal, but I agree that was not so clear for the message of this Table. Therefore, I redesigned it. It is most more clear in this way.